# Vascular dysfunction is associated with major adverse cardiovascular events in prediabetes: A cohort study

Dariya Kozlova[1☉], Colin Gimblet[1☉], Linder Wendt[2], Sadaf Akbari[1], Adeyinka Taiwo[1], Sanjana Dayal[1], Patrick Ten Eyck[2], Anna Stanhewicz[3], Joel Trinity[4], Diana Jalal[1,5*]

1 Department of Internal Medicine, Carver College of Medicine, University of Iowa, Iowa City, Iowa, United States of America, 2 Institute for Clinical and Translational Science, University of Iowa, Iowa City, Iowa, United States of America, 3 Department of Health and Human Physiology, University of Iowa, Iowa City, Iowa, United States of America, 4 Department of Medicine, University of Utah, Salt Lake City, Utah, United States of America, 5 Iowa City VA HCS, Iowa City, Iowa, United States of America

☉ These authors contributed equally to this work.
* diana-jalal@uiowa.edu

## Abstract

### Background

Prediabetes is a growing public health concern that increases the risk of major adverse cardiovascular events (MACE). Vascular dysfunction worsens with hyperglycemia and is associated with MACE in several high-risk populations. However, it is unknown whether vascular dysfunction contributes to MACE in prediabetes. We hypothesized that vascular dysfunction is associated with elevated risk of MACE in prediabetes.

### Methods

We conducted an observational study of 5742 adults (age 54.9 ± 11.5 years, 54% female) in the Framingham Offspring and Generation III cohorts. Prediabetes was defined using the ADA criteria. Endothelial function was determined via brachial artery flow-mediated dilation (BA-FMD), aortic stiffness via carotid-femoral pulse wave velocity (cfPWV), and coronary artery calcium (CAC) score via computed tomography. Stepwise selection models evaluated BA-FMD, cfPWV, and CAC score by prediabetes status. The association of BA-FMD, cfPWV, and CAC score with time to MACE was assessed via Cox proportional hazards regression.

### Results

Individuals with prediabetes had lower BA-FMD and higher cfPWV and CAC score (p < 0.001). In stepwise selection models, age, sex, smoking history, systolic blood pressure, triglycerides, high-density lipoprotein, low-density lipoprotein, and fasting glucose related to vascular dysfunction. After adjusting for traditional cardiovascular

**Data availability statement:** The manuscript does not contain the raw data. To obtain access to the data, investigators should submit requests for the data through BioLINCC, including an abstract, protocol/analysis plan, and IRB approval (expedited). Subsequently, a data use agreement will be completed prior to the release of the de-identified data. The data was made available to our group at no cost once these procedures were followed through: https://biolincc.nhlbi.nih.gov/home/

**Funding:** This study was supported by grants from the National Institutes of Health (HL134738 for D.J., T32 DK112751 for C.G., and HL169201 and HL168630 for A.S.), the National Center for Advancing Translational Sciences of the National Institutes of Health (UM1TR004403), as well as the Office of Research and Development and Department of Veterans Affairs (I01BX007087-06) to S.D. The funders had no role in study design, data collection and analysis, decision to publish, or preparation of the manuscript.

**Competing interests:** The authors have declared that no competing interests exist.

risk factors, BA-FMD (HR [95% CI], 0.93 [0.90,0.97]; p < 0.001) and CAC score >100 [HR [95% CI], 4.15 [2.24, 7.70]; p < 0.001)] were associated with MACE in prediabetes while cfPWV was not (p = 0.051).

## Conclusions

Vascular dysfunction measured by BA-FMD independently associates with MACE in prediabetes. Therapies that target vascular dysfunction may reduce CVD risk in prediabetes.

## Introduction

Prediabetes is a state of impaired glucose metabolism between normoglycemia and diabetes [1]. The prevalence of prediabetes continues to rise, affecting approximately 12.3 to 43.5% of adults in the United States based on the American Diabetes Association (ADA) criteria [2,3]. Notably, prediabetes increases the risk of all-cause mortality and cardiovascular disease (CVD) [3,4]. However, even after adjusting for traditional cardiovascular risk factors, established in the Framingham Heart Study, residual CVD risk persists [5–7]. Therefore, further research is needed to identify additional risk factors that contribute to CVD risk in individuals with prediabetes and to guide targeted therapies that reduce vascular complications.

Vascular dysfunction is increasingly recognized as a significant non-traditional risk factor that contributes to the elevated CVD risk in several populations. Specifically, reductions in vascular endothelial function assessed with brachial artery flow-mediated dilation (BA-FMD), increases in arterial stiffness measured via carotid-femoral pulse wave velocity (cfPWV), and higher coronary artery calcium (CAC) score are associated with cardiovascular events and all-cause mortality in several high-risk populations [8–13]. Individuals with prediabetes exhibit macrovascular and microvascular dysfunction and measures of vascular function including BA-FMD, cfPWV, and CAC score are known to worsen as blood glucose levels rise [14–19]. However, whether these measures of vascular function are associated with major adverse cardiovascular events (MACE) in prediabetes is unknown. This study aimed to assess the relation between BA-FMD, cfPWV, and CAC score, as measures of vascular dysfunction, with future MACE in adults with prediabetes compared to normoglycemic adults. We hypothesized that lower BA-FMD and higher cfPWV and CAC score would be associated with a higher risk of MACE in individuals with prediabetes, independently of traditional CVD risk factors.

## Materials and methods

### Study design

Data collected between 1998–2019 from the Framingham Offspring Study and Framingham Generation 3 cohorts were included in this analysis (n = 5742). Detailed descriptions of the Framingham cohorts are available elsewhere [20–22]. This study was approved by the University of Iowa Human Subjects Office and Institutional

Review Board (IRB 201811756). As de-identified data was obtained from BIOLINCC, a waiver of informed consent was provided. Inclusion criteria for this analysis comprised of subjects with a hemoglobin A1c (HbA1c) or fasting glucose measurements available at the baseline visit. For the Framingham Offspring cohort, the baseline visit was considered exam 7, and data was included for all the subsequent visits through exam 9. In the Framingham Generation 3 cohort, the baseline visit was considered exam 2, with a follow-up visit at exam 3. Participants were excluded from the analysis if they had diabetes at baseline. Diabetes was defined in the Framingham data set as blood glucose at exam ≥ 200 mg/dL or fasting blood glucose ≥126 mg/dL or current treatment for diabetes at exam visit. Prediabetes was defined by the ADA fasting glucose criteria (fasting plasma glucose between 100–125 mg/dL) or meeting ADA HbA1c criteria (HbA1c between 5.7–6.4% (39–46 mmol/mol)) [1]. Otherwise, participants were classified as normoglycemic.

## Clinical characteristics and outcome measures

Participant age, sex, race, body mass index (BMI), history of CVD, estimated glomerular filtration rate (eGFR), and smoking status were obtained from baseline visits within the Framingham Offspring Study and Framingham Generation 3 cohorts. Additional vitals and laboratory measures including HbA1c, fasting glucose, systolic blood pressure, diastolic blood pressure, triglycerides, low-density lipoprotein (LDL), high-density lipoprotein (HDL), and total cholesterol were also obtained from baseline Framingham visits. Participants missing baseline clinical characteristics were not excluded from analysis unless both baseline fasting glucose and HbA1c values were missing. Age, systolic blood pressure, LDL, HDL, total cholesterol, and fasting glucose were reported in increments of 10 for analyses. Triglycerides were reported in increments of 50 for analysis. eGFR was calculated via the 2021 CKD-EPI creatinine equation, and chronic kidney disease (CKD) was defined as eGFR < 60 at baseline visit [23].

Endothelial function was determined via BA-FMD, aortic stiffness via cfPWV, and CAC score via computed tomography CAC score classified by Agatston score as previously described [24–26]. CAC score was then grouped into the following categories: 0, > 0–100, > 100. MACE was defined as incident CHD, stroke, and all-cause mortality after the defined baseline visit [27]. Participants with a history of CVD were excluded from time-to-event analyses evaluating MACE. Only participants with prediabetes were included in the evaluation of MACE. Participants were not excluded from the analysis if they were lost to follow up.

## Statistical analysis

Summary statistics were reported according to prediabetes vs normoglycemia as median (interquartile range) for continuous variables and as number (percentage of participants) for categorical variables. P-values were obtained using the Wilcoxon rank sum test and Pearson's Chi-squared test. Forward stepwise selection based on the Akaike information criterion (AIC) was then used to identify optimal model predictor sets for the outcome measures of BA-FMD, cfPWV, and CAC score in the participants with prediabetes. Stepwise models utilized transformations of the outcome variables as appropriate: BA-FMD models utilized standard linear modeling, cfPWV models utilized log-transformed Gamma modeling, and CAC score models utilized cumulative logistic modeling. Stepwise model selection used the following candidate variables: age, sex, history of CVD, CKD, smoking status, systolic blood pressure, diastolic blood pressure, BMI, A1c, fasting glucose, total cholesterol, triglycerides, LDL, HDL, creatinine, and eGFR. For each outcome, final model point estimates (Mean Difference, Mean Ratio, Odds Ratio), confidence intervals, and p-values were generated using complete-case samples based on the variables in the optimal model according to AIC. Subsequently, we evaluated if BA-FMD, cfPWV, and CAC score predicted MACE. A log-rank test was used to evaluate MACE events over time between participants with prediabetes and normoglycemic participants. Notably, participants with a history of CVD were excluded from this portion of the analysis. Cox proportional hazards modeling was used to estimate and test the associations of BA-FMD, cfPWV, and CAC score with time to MACE in prediabetes adjusting for age, sex, smoking status, eGFR, BMI, systolic blood pressure, and LDL. Interaction terms for age, sex, or prediabetes definition and vascular function or CAC were evaluated in

an additional set of models. Hazard ratios and their corresponding confidence intervals and p-values were provided. The proportional hazards assumption was evaluated using Schoenfeld residuals. Due to the robust sample size of our study, a more stringent cutoff of p = 0.01 was used in this context, with Schoenfeld residual p-values greater than 0.01 signifying that Cox proportional hazards modeling was an acceptable framework. Significance was set at an alpha level of 0.05 for all other analyses.

## Results

### Participant characteristics

Demographics of the study cohort are shown in Table 1. Among participants with prediabetes, 48% met the ADA fasting glucose criteria for prediabetes, 27% met the ADA HbA1c criteria, and 25% met the diagnostic prediabetes criteria based on both fasting glucose and HbA1c criteria. Median follow up time was 13.2 years (IQR 12.2–14.1). Compared to those in the normoglycemia group, participants with prediabetes were older and a greater proportion were male (p < 0.001). Additionally, most participants were white. A greater number of participants in the prediabetes group were current smokers and had a history of CVD (p < 0.001). Furthermore, body mass index (BMI), systolic blood pressure, diastolic blood pressure, triglycerides, LDL, and total cholesterol were higher in the prediabetes group (p < 0.001). Conversely, HDL was lower in the prediabetes group (p < 0.001). Both groups had normal median kidney function according to KDIGO guidelines although CKD was more common in the prediabetes group (p < 0.001) [23].

### Vascular function and CAC score in prediabetes

Individuals with prediabetes had lower BA-FMD compared with the normoglycemia group, 3.4% (1.4, 5.9) and 4.7% (2.3, 7.4), respectively (p < 0.001; Fig 1A). Additionally, individuals with prediabetes also had a higher median cfPWV of 8.09 m/s (7.00, 9.83) compared with individuals in the normoglycemia group who had a median cfPWV of 6.95 m/s (6.17, 8.15) (p < 0.001; Fig 1B). A greater proportion of individuals with prediabetes had a CAC score greater than zero (p < 0.001; Fig 1C). As shown in Fig 1C, 60% of participants in the normoglycemia group had a CAC score of 0, 22% had a CAC score of >0–100, and 18% had a CAC score >100, while in the prediabetes group, 44% had a CAC score of 0, 23% had a CAC score of >0–100, and 33% had a CAC score >100.

### Variables associated with vascular function in prediabetes

Multivariable models determined via stepwise selection among participants with prediabetes are shown in Table 2. Advancing age (p < 0.001) and elevated systolic blood pressure (p < 0.001) were associated with lower BA-FMD. Female sex (p < 0.001) and greater HDL (p = 0.05) were associated with higher BA-FMD. The following variables were associated with higher cfPWV: advancing age (p < 0.001), elevated systolic blood pressure (p < 0.001), higher triglycerides (p = 0.002), and greater fasting glucose (p = 0.032). Former smoking status (p = 0.043) and greater HDL (p = 0.012) were associated with lower cfPWV. In addition, the following variables were associated with higher CAC score: advancing age (OR [95% CI], 4.79 [3.83, 6.05]; p < 0.001), male sex (p < 0.001), former smoking status (p < 0.001), current smoking status (p = 0.002), and higher total cholesterol (p = 0.001). Higher HDL was associated with a lower CAC score (p < 0.001).

### MACE in prediabetes

Individuals in the prediabetes group experienced a greater prevalence of MACE (30%) compared with those in the normoglycemia category (15%) (p < 0.001). The incidence rate of MACE events in the overall population was 21%. The median time-to-event/loss to follow-up for participants was 12.7 years for the prediabetes group, 13.4 years for the normoglycemia group and 13.1 years for the two groups combined. Participants with prediabetes were at an increased risk of MACE over time compared to normoglycemic participants (HR [95% CI], 2.34 [2.02, 2.72]; p < 0.001). The association of prediabetes

**Table 1. Clinical characteristics.**

| | Normoglycemia (n = 3314) | Prediabetes (n = 2428) |
|---|---|---|
| Age, years | 49 (42, 57) | 56 (49, 64) |
| Sex | | |
| Male | 1269 (38%) | 1353 (56%) |
| Female | 2045 (62%) | 1075 (44%) |
| Race | | |
| White | 3155 (100%) | 2203 (99%) |
| Missing | 203 | 146 |
| History of Cardiovascular Disease, % | 94 (2.8%) | 152 (6.3%) |
| Missing | 0 | 0 |
| CKD* | 54 (1.6%) | 91 (3.8%) |
| Missing | 6 | 3 |
| Smoking Status | | |
| Never Smoker | 1760 (53%) | 1037 (43%) |
| Former Smoker | 1147 (35%) | 1012 (42%) |
| Current Smoker | 407 (12%) | 379 (16%) |
| SBP, mmHg | 115 (106, 125) | 123 (114, 135) |
| Missing | 0 | 0 |
| DBP, mmHg | 73 (67, 79) | 76 (69, 82) |
| Missing | 0 | 2 |
| eGFR, ml/min/1.73m$^2$ | 97 (86, 106) | 93 (81, 102) |
| Missing | 6 | 3 |
| BMI, kg/m$^2$ | 25.9 (23.1, 29.0) | 28.6 (25.6, 32.2) |
| Missing | 3 | 4 |
| Triglycerides, mg/dL | 90 (65, 129) | 116 (83, 163) |
| Missing | 0 | 0 |
| LDL, mg/dL | 106 (86, 128) | 116 (94, 137) |
| Missing | 31 | 17 |
| HDL, mg/dL | 59 (48, 71) | 51 (41, 62) |
| Missing | 5 | 1 |
| Total Cholesterol, mg/dL | 188 (168, 212) | 196 (172, 219) |
| Missing | 0 | 0 |
| Fasting Glucose, mg/dL | 90 (86, 94) | 101 (96, 106) |
| Missing | 1,333 | 1,231 |
| HbA1c, % | 5.30 (5.10, 5.49) | 5.70 (5.40, 5.90) |
| mmol/mol | 34 (32, 36) | 39 (36, 41) |
| Missing | 86 | 0 |
| Prediabetes Criteria Met | | |
| Fasting Glucose | | 1113 (48%) |
| HbA1c | | 638 (27%) |
| Fasting Glucose & HbA1c | | 572 (25%) |

Variables are presented as median (interquartile range) for continuous variables and number (percentage of participants) for categorical variables. Wilcoxon rank sum test and Pearson's Chi-squared test were used to evaluate for significant differences between the normoglycemia and prediabetes groups. $p \le 0.05$. Chronic kidney disease (CKD), systolic blood pressure (SBP); diastolic blood pressure (DBP); estimated glomerular filtration rate (eGFR); low-density lipoprotein (LDL); high-density lipoprotein (HDL).

*CKD was defined as eGFR < 60 mL/min/1.73m$^2$.

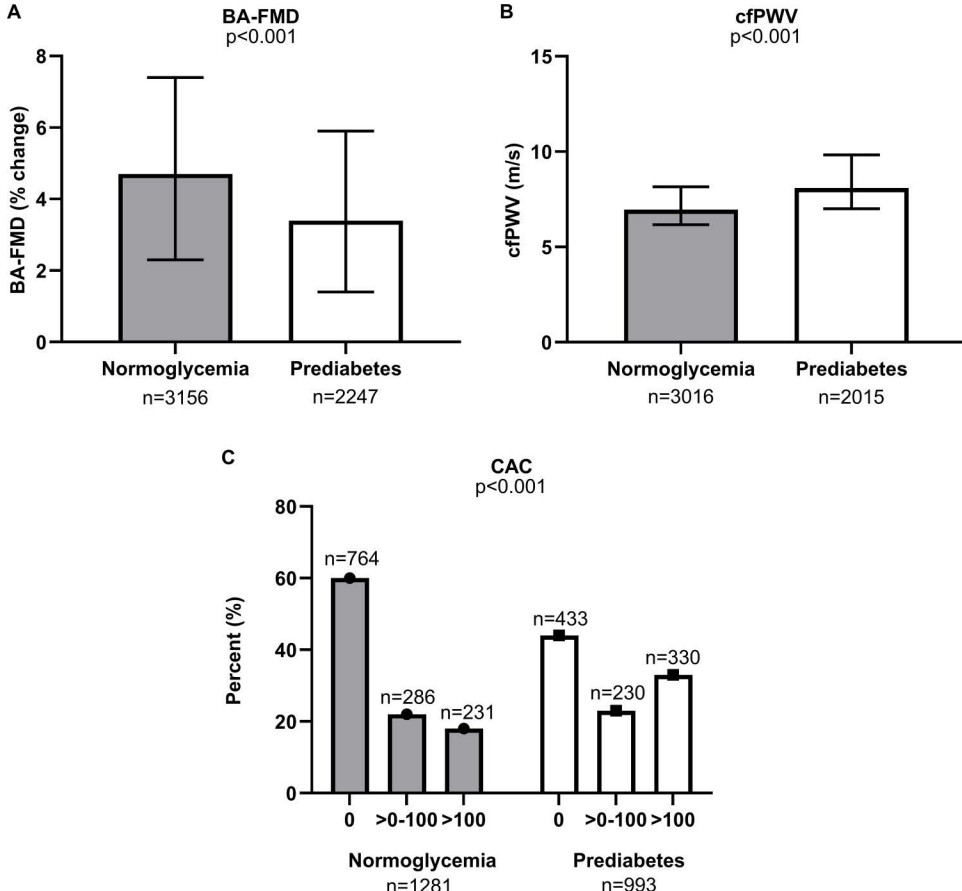

**Fig 1. Vascular function and CAC score in Framingham.** A. BA-FMD in normoglycemic participants and participants with prediabetes in the Framingham cohort. Data reported as median and interquartile range. p ≤ 0.05 B. cfPWV in normoglycemic participants and participants with prediabetes in the Framingham cohort. Data reported as median and interquartile range. p ≤ 0.05 C. CAC score in normoglycemic participants and participants with prediabetes in the Framingham cohort. Data is reported as the percentage of participants per group. p ≤ 0.05 All figure p values reported refer to differences in normoglycemia and prediabetes groups.

with MACE was no longer significant after adjusting for age, sex, smoking, BMI, systolic blood pressure, and LDL (HR [95% CI], 1.09 [0.93, 1.28]; p = 0.3).

Table 3 shows unadjusted and adjusted associations of BA-FMD, cfPWV, and CAC score with MACE in prediabetes. Higher BA-FMD was associated with an 18% lower risk of MACE. Additionally, higher cfPWV was associated with a 16% higher risk of MACE. Furthermore, a CAC score of >0–100 was associated with a 173% greater risk of MACE, and a CAC score of >100 was associated with a 11.3 times higher risk of MACE (HR [95% CI], 11.3 [6.71,19.0]; p < 0.001). After adjustment for age, sex, smoking status, systolic blood pressure, BMI, and LDL, BA-FMD and CAC score > 100 remained significantly associated with MACE (Table 3). Additional adjustment for fasting glucose or the inclusion of interaction terms for age, sex, or prediabetes definition did not attenuate this relation. All time-to-event models, both adjusted and unadjusted, had Schoenfeld residual p-values greater than 0.01.

## Discussion

Our study aimed to evaluate the degree of vascular dysfunction in prediabetes and assess whether vascular dysfunction was associated with MACE in prediabetes. We observed that individuals with prediabetes exhibit marked vascular

**Table 2. Stepwise selection models of BA-FMD, cfPWV, and CAC score in participants with prediabetes.**

| Variable | BA-FMD (n = 2247) R²=0.230 Mean Diff (95% CI) | P value | cfPWV (n = 2015) R²=0.651 Mean Ratio (95% CI) | P value | CAC (n = 993) Odds Ratio (95% CI) | P value |
|---|---|---|---|---|---|---|
| Age, years | −1.2 (−1.3, −1.0) | <0.001 | 1.15 (1.14, 1.17) | <0.001 | 4.79 (3.83, 6.05) | <0.001 |
| Female v Male | 1.5 (1.1, 1.8) | <0.001 | 0.98 (0.96, 1.00) | 0.068 | 0.33 (0.23, 0.46) | <0.001 |
| Smoking Status | | | | | | |
| Never Smoker | – | – | Ref | Ref | Ref | Ref |
| Former Smoker | – | – | 0.98 (0.96, 1.00) | 0.043 | 2.17 (1.59, 2.99) | <0.001 |
| Current Smoker | – | – | 0.98 (0.96, 1.01) | 0.2 | 2.19 (1.35, 3.57) | 0.002 |
| SBP, mmHg | −0.28 (−0.39, −0.18) | <0.001 | 1.06 (1.05, 1.06) | <0.001 | – | – |
| Triglycerides, mg/dL | – | – | 1.01 (1.00, 1.02) | 0.002 | – | – |
| HDL, mg/dL | 0.10 (0.00, 0.20) | 0.050 | 0.99 (0.98, 1.00) | 0.012 | 0.74 (0.66, 0.82) | <0.001 |
| Total cholesterol, mg/dL | – | – | – | – | 1.07 (1.03, 1.12) | 0.001 |
| Fasting Glucose, mg/dL | – | – | 1.01 (1.00,1.02) | 0.032 | 1.16 (0.97, 1.40) | 0.10 |

BA-FMD models utilized standard cross sectional linear modeling, cfPWV models utilized log-transformed Gamma modeling, and CAC score models utilized cumulative logistic modeling. Variables included in the model: age, sex, history of cardiovascular disease, history of chronic kidney disease, smoking status, diastolic blood pressure, systolic blood pressure, BMI, A1c, creatinine, fasting glucose, total cholesterol, triglycerides, LDL, HDL, and eGFR. Age, blood pressure, HDL, and fasting glucose were reported in 10-unit increments. Triglycerides were reported in 50-unit increments. The reference group for smoking status was designated as never smokers. p ≤ 0.05. Coronary artery calcium score (CAC); brachial artery flow-mediated dilation (BA-FMD); carotid-femoral pulse wave velocity (cfPWV); systolic blood pressure (SBP), diastolic blood pressure (DBP); estimated glomerular filtration rate (eGFR); low-density lipoprotein (LDL); high-density lipoprotein (HDL).

**Table 3. Associations of BA-FMD, cfPWV, and CAC score with incident MACE in prediabetes.**

| Variable | MACE – Unadjusted Hazard Ratio (95% CI) | P value | MACE – Adjusted* Hazard Ratio (95% CI) | P value |
|---|---|---|---|---|
| BA-FMD (n = 1925) | 0.82 (0.80, 0.85) | <0.001 | 0.93 (0.90, 0.97) | <0.001 |
| cfPWV (n = 1722) | 1.16 (1.14, 1.17) | <0.001 | 1.03 (1.00, 1.05) | 0.051 |
| CAC (n = 992) | | | | |
| 0 | Ref | Ref | Ref | Ref |
| >0-100 | 2.73 (1.45, 5.15) | 0.002 | 1.64 (0.84, 3.20) | 0.147 |
| >100 | 11.3 (6.71, 19.0) | <0.001 | 4.15 (2.24, 7.70) | <0.001 |

Models were adjusted for age, sex, smoking status, systolic blood pressure, eGFR, body mass index, and low-density lipoprotein. CAC score of zero was designated as the reference group for CAC score analysis. Blood pressure was reported in 10 mmHg increments. p ≤ 0.05. Brachial artery flow-mediated dilation (BA-FMD); carotid-femoral pulse wave velocity (cfPWV); coronary artery calcium score (CAC).

dysfunction. Additionally, endothelial dysfunction and CAC score > 100 were associated with future MACE in prediabetes after adjusting for confounding variables. These data indicate that these measures of vascular dysfunction predict MACE in individuals with prediabetes. Given that CVD risk in prediabetes is independent of whether patients progress to overt diabetes [4,5], interventions that improve vascular function may help mitigate CVD risk in this population.

Endothelial dysfunction, CAC score, and aortic stiffness are non-traditional risk factors that precede the development of CVD [1,28]. Previous studies have shown that vascular dysfunction is present in people with impaired glucose tolerance, however, the findings are inconsistent, partly due to variations in the criteria used to define prediabetes [14–16]. For instance, the Hoorn study found that individuals with impaired glucose tolerance had normal BA-FMD, while the FMD-J study showed lower BA-FMD in individuals with impaired glucose tolerance [14]. Prior studies of

impaired glucose tolerance often use an oral glucose tolerance test to define prediabetes, however, oral glucose tolerance tests are not commonly utilized in clinical practice [10,14,16]. In contrast, we used the ADA fasting glucose and HbA1c criteria to reflect clinical practice and found that BA-FMD was impaired in prediabetes. Additionally, individuals with prediabetes had greater coronary artery calcification determined with CAC scores. While previous studies have not reached a clear consensus on the association between prediabetes and CAC, often implicating BMI as a confounder [17,29], our results showed that including BMI did not improve the prediction of CAC scores in the prediabetes group. Instead, age emerged as a significant predictor in prediabetes, with CAC score increasing by 379% per decade. Furthermore, we observed that aortic stiffness, measured by cfPWV, was higher in individuals with prediabetes. This agrees with prior studies reporting higher cfPWV in participants with impaired glucose metabolism [16]. Taken together, our data suggest that prediabetes is characterized by conduit artery endothelial dysfunction, calcification of coronary arteries, and reduced elasticity of the aorta.

Extensive research has established that prediabetes significantly increases the risk of CVD [4,5,30,31]. This agrees with our finding that prediabetes was associated with a higher incidence of MACE compared with normoglycemic participants, although our observed association was attenuated after adjusting for traditional CVD risk factors. Importantly, few studies have evaluated the association between vascular dysfunction and MACE in prediabetes. Cruickshank et al. found that aortic stiffness was associated with mortality in individuals with impaired glucose tolerance [10]. Our study expands upon these findings by demonstrating that aortic stiffness was not associated with MACE in prediabetes after adjusting for traditional cardiovascular risk factors in our cohort. Additionally, to our knowledge, this is the first study to demonstrate that BA-FMD and CAC score >100 are independently associated with MACE in prediabetes, beyond traditional CVD risk factors. Of note, the addition of risk factors significantly attenuated this relationship, though significance remained. Overall, our study implicates vascular dysfunction as a novel, independent risk factor that increases the risk of MACE in prediabetes.

Several mechanisms likely contribute to vascular dysfunction in prediabetes, particularly with aging. In an observational study of healthy older adults, vascular dysfunction was associated with cardiovascular risk factors such as smoking, elevated blood pressure, and hyperlipidemia [32–35]. Similarly, we observed that age, smoking, and metabolic syndrome risk factors contribute to vascular dysfunction in prediabetes. Notably, endothelial dysfunction is characterized by a reduction in nitric oxide bioavailability, generated by the enzyme endothelial nitric oxide synthase (eNOS) [36]. Hyperglycemia in prediabetes leads to endothelial dysfunction by increasing the production of reactive oxygen species (ROS), such as superoxide, within mitochondria [37]. Once produced, ROS can uncouple eNOS, causing eNOS to produce superoxide instead of nitric oxide [36]. In coronary arteries, endothelial dysfunction allows LDL to migrate into the subintimal space and interact with ROS, leading to the oxidation of LDL and coronary artery calcification [38]. In the medial layer of the aorta, hyperglycemia promotes advanced glycation end-product formation and ROS production, resulting in elastin fragmentation, collagen deposition, and cross-linking of elastin and collagen, thereby accelerating vascular stiffening [39,40]. Collectively, traditional CVD risk factors and hyperglycemia-induced ROS production contribute to vascular dysfunction in prediabetes.

Interventions that target cardiovascular risk factors in adults with prediabetes, such as exercise, smoking cessation, and other lifestyle modifications in addition to pharmacologic therapies, may enhance vascular function in prediabetes allowing for early intervention (prior to the onset of overt diabetes and clinical CVD). The association of measures of vascular dysfunction with MACE in prediabetes suggest that the utilization of BA-FMD or CAC as surrogate outcomes may be appropriate to evaluate potential interventions in this patient population. Of note, interventions that improve vascular dysfunction have been shown to impact hard clinical outcomes such as cardiovascular events and/or mortality in other populations [41–46]. A key strength of our study is the use of the Framingham Heart Study cohort, a well-characterized, longitudinal population study with standardized data collection, well-defined defined MACE, and robust vascular function assessments.

There were several limitations to our study. First, our participants were predominantly white. The 2024 ADA Standards of Care in Diabetes recommend screening for prediabetes in asymptomatic adults based on several risk factors, including race [1]. Since our study population does not reflect the full demographic diversity of the United States, our findings may not fully capture the CVD risk across different racial and ethnic groups. CVD risk is known to be higher in minority groups and both CVD risk and vascular function are impacted by access to social supports promoting exercise and other lifestyle modifications [16,47,48]. Additionally, vascular function is dramatically reduced in post-menopausal women [49]. We did not characterize menopause status in our study and therefore cannot evaluate whether vascular dysfunction has a greater contribution to MACE in pre-menopausal or post-menopausal women with prediabetes. Future studies could stratify participants by menopausal status to further evaluate the effects of vascular function on MACE risk in pre-menopausal and post-menopausal women. Our study also did not evaluate the impact of lifestyle modifications such as exercise or diet, and environmental factors on the prediction of MACE using vascular function in prediabetes. These factors are known to influence vascular function and therefore may influence this relation. Lastly, our study utilized the ADA fasting glucose and HbA1c definition of prediabetes, resulting in a study sample that primarily fell within the lower spectrum of prediabetes range hyperglycemia. Although this approach may not capture individuals at the highest risk of MACE, our findings demonstrate that impaired glucose metabolism remains a significant contributor to vascular dysfunction, even when applying a broadly defined clinical prediabetes criteria.

In conclusion, individuals with prediabetes display vascular dysfunction, which is independently associated with MACE. Future research may evaluate potential therapies that improve cardiovascular risk in prediabetes. Further research into therapies that enhance vascular function in prediabetes may offer a viable approach to lowering CVD risk in this population.

## Author contributions

**Conceptualization:** Dariya Kozlova, Colin Gimblet, Sanjana Dayal, Joel Trinity, Diana Jalal.

**Data curation:** Dariya Kozlova.

**Formal analysis:** Linder Wendt, Patrick Ten Eyck, Diana Jalal.

**Investigation:** Dariya Kozlova, Colin Gimblet, Sadaf Akbari, Adeyinka Taiwo, Anna Stanhewicz, Joel Trinity, Diana Jalal.

**Methodology:** Colin Gimblet, Linder Wendt.

**Project administration:** Diana Jalal.

**Supervision:** Colin Gimblet, Diana Jalal.

**Writing – original draft:** Dariya Kozlova.

**Writing – review & editing:** Colin Gimblet, Sadaf Akbari, Adeyinka Taiwo, Sanjana Dayal, Patrick Ten Eyck, Anna Stanhewicz, Joel Trinity, Diana Jalal.

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
