## [Decision Letter · Decision Letter 0]

27 Jan 2025

PONE-D-24-59079Vascular dysfunction is associated with major adverse cardiovascular events in prediabetesPLOS ONE

Dear Dr. Jalal,

Thank you for submitting your manuscript to PLOS ONE. After careful consideration, we feel that it has merit but does not fully meet PLOS ONE’s publication criteria as it currently stands. Therefore, we invite you to submit a revised version of the manuscript that addresses the points raised during the review process.

We look forward to receiving your revised manuscript.

Kind regards,

Neftali Eduardo Antonio-Villa, MD PhD

Academic Editor

PLOS ONE

Journal Requirements:

“Nothing to disclose”

4. Please note that your Data Availability Statement is currently missing the repository name and/or the DOI/accession number of each dataset OR a direct link to access each database. If your manuscript is accepted for publication, you will be asked to provide these details on a very short timeline. We therefore suggest that you provide this information now, though we will not hold up the peer review process if you are unable.

Additional Editor Comments:

Jalal Diana et al. performed a subanalysis of the Framingham Offspring Study and the Generation III cohort to evaluate differences in vascular dysfunction parameters (brachial artery flow-mediated dilation (BA-FMD), aortic stiffness via carotid-femoral pulse wave velocity (cfPWV), and coronary artery calcium (CAC)), predictors of these conditions, and whether vascular dysfunction predicts incident MACE events. The authors found that patients with prediabetes exhibited impaired vascular hemodynamics, tested several predictors of these conditions, and concluded that BA-FMD and CAC predict MACE events. This is a well-structured study with potential implications for diabetes care. However, together with three external referees, we identified several concerns that should be addressed before a final decision on this manuscript is made. Below are my suggestions:

• In the abstract, please include the mean age and the proportions of women and men as part of the descriptive characteristics of the studied sample.

• The objective is stated in a very broad and untargeted way. I recommend the authors restructure the study aims using the PECOT format to specifically mention the primary and secondary objectives.

• Please include the STROBE checklist for observational studies as a supplementary appendix.

• I suggest rephrasing “history of CKD” to “decreased eGFR” to ensure precision in describing this metric.

• Please justify the rationale for grouping the CAC scores into 0, 1–100, and ≥100. Were patients with CAC scores between 0 and 1 included in the study?

• Include a reference to justify the choice of incident MACE events as an outcome. Most MACE definitions exclude “all-cause mortality” and instead focus on “CVD-related mortality.”

• Did the authors encounter missing data? This is not specified in the statistical plan or the methods section.

• Although the AIC method is a valid statistical technique to identify predictors, it is often untargeted and may not yield the best physiopathological insights. I strongly recommend that the authors adopt a more structured approach, such as using directed acyclic graphs (DAGs), to test and support their predictors.

• Please include the R² values for the models used to evaluate predictors of vascular dysfunction.

• Regarding the Cox regression models, did the authors test the proportional hazards assumption? This can be evaluated using Schoenfeld residuals.

• Consider other potential confounders in the studied relationships, such as BMI, eGFR, and A1C. These could be included in an additional model, even as part of a sensitivity analysis.

• Please include the median follow-up time and the interquartile range (IQR). Furthermore, estimate the incidence rate of MACE events in the studied population.

• It would be valuable if the authors tested whether different prediabetes definition criteria (e.g., fasting glucose vs. normoglycemia, HbA1c only vs. normoglycemia, and both vs. normoglycemia) individually predict MACE outcomes. This information would be particularly useful for clinicians, especially in low-resource settings where HbA1c is not always available.

• The authors should expand on the clinical relevance of their findings. How can clinicians evaluate and address vascular dysfunction in clinical practice to mitigate the risk of adverse cardiometabolic outcomes in patients with prediabetes?

• Finally, consider the potential impact of residual confounding on the results. Factors such as lifestyle, diet, and environmental exposures, which influence both vascular dysfunction and MACE events, should be discussed.

Reviewers' comments:

Reviewer's Responses to Questions

**Comments to the Author**

1. Is the manuscript technically sound, and do the data support the conclusions?

Reviewer #1: Yes

Reviewer #2: Yes

Reviewer #3: Yes

2. Has the statistical analysis been performed appropriately and rigorously? 

Reviewer #1: Yes

Reviewer #2: Yes

Reviewer #3: Yes

3. Have the authors made all data underlying the findings in their manuscript fully available?

Reviewer #1: Yes

Reviewer #2: Yes

Reviewer #3: Yes

4. Is the manuscript presented in an intelligible fashion and written in standard English?

Reviewer #1: Yes

Reviewer #2: Yes

Reviewer #3: Yes

5. Review Comments to the Author

Reviewer #1: Abstract: which CVD risk factors

Introduction: what does a non-traditional risk factor mean? Versus traditional?

Results:

1. The observed association between prediabetes and MACE was attenuated after adjusting for traditional risk factors, which was attributed to sample size limitations. How could the limited sample size have affected the findings, did power calculations suggest insufficient statistical power?

2. You mention that their sample was predominantly white and that this may limit generalizability. Could you comment on the potential impact of race and ethnicity on vascular dysfunction and MACE risk in prediabetes, perhaps referencing studies that have shown different cardiovascular risk profiles across racial groups?

3. Given that there is the limitation of not including menopause status in their study, which is important due to the known effects of menopause on vascular health. It would be helpful if you could suggest how future studies could account for hormonal factors or explore this further in prediabetic women. For example, a follow-up study could stratify by menopause status or investigate hormonal therapy as a potential modulator of vascular dysfunction.

4. I might suggest that the authors discuss how their results could influence existing treatment guidelines for prediabetes and CVD prevention. Would interventions targeting vascular dysfunction be integrated into current prediabetes management strategies, or is further evidence needed before such recommendations are made?

5. The paper could also expand on the potential for early interventions. For example, could lifestyle interventions like exercise or dietary changes have a meaningful impact on vascular function in people with prediabetes before they progress to full diabetes?

Reviewer #2: Dear authors,

Thank you for the your effort and contribution to the field with this study. Your work on a topic of such clinical importance is appreciated. Below are some suggestions to enhance the clarity and rigor of the manuscript:

1. Although the authors mention the Framingham cohorts later in the text, they fail to justify why this dataset is particularly well-suited to address their research question. The manuscript would benefit greatly from a compelling rationale for selecting these cohorts. Why should the reader consider the use of this cohort as a significant strength of the study>

2. The authors state that data are available through BioLINCC but fail to provide a direct link or accession details. This lack of transparency makes it challenging for other researchers to verify or reproduce the results.

3. The Results section does not address potential interaction effects between key variables, such as age, sex, or prediabetes criteria and vascular dysfunction. This is a significant oversight in a study aiming to examine independent associations. Were vascular dysfunction measures equally predictive of MACE across subgroups?

4. The dramatic reduction in the hazard ratios, such as for CAC > 100 (from 11.3 unadjusted to 3.90 after adjustment) is not adequately explained. This substantial attenuation suggests major confounding factors, yet the authors fail to critically examine or discuss this.

5. The authors imply causation between vascular dysfunction and MACE in prediabetes, despite their study being observational in design. For example, they state: “Our findings suggest that the elevated CVD risk observed in prediabetes may be mediated, in part, by vascular dysfunction” Such claims must be tempered, with explicit acknowledgment of the limitations of causal inference in the context of Cox regression analysis.

6. While the authors suggest that therapies targeting vascular dysfunction may reduce MACE in prediabetes, they do not provide specific recommendations or actionable insights. How do their findings translate into clinical practice? What are the next steps for researchers and clinicians? Providing clarity here would enhance the utility of the study.

7. Providing a legend for the figure would help readers interpret the figures more effectively.

Reviewer #3: Research has proven that prediabetes is a risk for CVD. Vascular dysfunction is a non-traditional risk factor that increases CVD risk. Measures of vascular function such as BA-FMD, cfPWV, and CAC score have been associated with an elevated risk of cardiovascular events and all-cause mortality. The authors, therefore, conducted an observational study to assess the relationship between vascular dysfunction and the risk of MACE in prediabetes. The study design and tests align with the study's objectives. The results were presented logically and are sufficient to conclude that vascular dysfunction is independently associated with MACE in prediabetes.

Suggestions

Results section

I suggest that the authors include subheadings for the results so the reader can easily follow and understand the order of the results e.g, a subheading before line 178 where the authors present results on BA-FMD, cfPWV and CAC score. A subheading before line 197

6. PLOS authors have the option to publish the peer review history of their article (what does this mean? ). If published, this will include your full peer review and any attached files.

**Do you want your identity to be public for this peer review?** For information about this choice, including consent withdrawal, please see our Privacy Policy .

Reviewer #1: No

Reviewer #2: **Yes: ** Afshin Heidari, MD

Reviewer #3: No

---

## [Author Response · Author response to Decision Letter 1]

31 Mar 2025

Please see attached Response to the Reviewers document

---

## [Decision Letter · Decision Letter 1]

4 May 2025

Vascular dysfunction is associated with major adverse cardiovascular events in prediabetes

PONE-D-24-59079R1

Dear Dr. Jalal,

We’re pleased to inform you that your manuscript has been judged scientifically suitable for publication and will be formally accepted for publication once it meets all outstanding technical requirements.

Kind regards,

Neftali Eduardo Antonio-Villa, MD PhD

Academic Editor

PLOS ONE

Additional Editor Comments (optional):

Dear authors, Thank you for addressing the comments made by all reviewers. The manuscript is suitable for publication in Plos One

Reviewers' comments:

Reviewer's Responses to Questions

**Comments to the Author**

1. If the authors have adequately addressed your comments raised in a previous round of review and you feel that this manuscript is now acceptable for publication, you may indicate that here to bypass the “Comments to the Author” section, enter your conflict of interest statement in the “Confidential to Editor” section, and submit your "Accept" recommendation.

Reviewer #3: All comments have been addressed

2. Is the manuscript technically sound, and do the data support the conclusions?

Reviewer #3: Yes

3. Has the statistical analysis been performed appropriately and rigorously? 

Reviewer #3: Yes

4. Have the authors made all data underlying the findings in their manuscript fully available?

Reviewer #3: Yes

5. Is the manuscript presented in an intelligible fashion and written in standard English?

Reviewer #3: Yes

6. Review Comments to the Author

Reviewer #3: (No Response)

7. PLOS authors have the option to publish the peer review history of their article (what does this mean? ). If published, this will include your full peer review and any attached files.

**Do you want your identity to be public for this peer review?** For information about this choice, including consent withdrawal, please see our Privacy Policy .

Reviewer #3: No

---

## [Editor Report · Acceptance letter]

PONE-D-24-59079R1

PLOS ONE

Dear Dr. Jalal,

I'm pleased to inform you that your manuscript has been deemed suitable for publication in PLOS ONE. Congratulations! Your manuscript is now being handed over to our production team.

Kind regards,

on behalf of

Dr. Neftali Eduardo Antonio-Villa

Academic Editor

PLOS ONE